# Exploring the Potential of Exosomal Biomarkers in Mild Traumatic Brain Injury and Post-Concussion Syndrome: A Systematic Review

**DOI:** 10.3390/jpm14010035

**Published:** 2023-12-27

**Authors:** Ioannis Mavroudis, Sidra Jabeen, Ioana Miruna Balmus, Alin Ciobica, Vasile Burlui, Laura Romila, Alin Iordache

**Affiliations:** 1Department of Neuroscience, Leeds Teaching Hospitals, NHS Trust, Leeds LS2 9JT, UK; i.mavroudis@nhs.net; 2Liaquat National Hospital and Medical College, Karachi 74800, Pakistan; 3Department of Exact Sciences and Natural Sciences, Institute of Interdisciplinary Research, “Alexandru Ioan Cuza” University of Iasi, 26th Alexandru Lapusneanu Street, 700057 Iasi, Romania; 4Department of Biology, Faculty of Biology, Alexandru Ioan Cuza University of Iasi, 20th Carol I Avenue, 700506 Iași, Romania; 5Preclinical Department, Apollonia University, Păcurari Street 11, 700511 Iasi, Romania; 6Faculty of Medicine, University of Medicine and Pharmacy “Grigore T. Popa”, 700115 Iasi, Romania; aliniordache@yahoo.com

**Keywords:** mild traumatic brain injury, post-concussion syndrome, exosomes, salivary biomarkers

## Abstract

Background: Alongside their long-term effects, post-concussion syndrome (PCS) and mild traumatic brain injuries (mTBI) are significant public health concerns. Currently, there is a lack of reliable biomarkers for diagnosing and monitoring mTBI and PCS. Exosomes are small extracellular vesicles secreted by cells that have recently emerged as a potential source of biomarkers for mTBI and PCS due to their ability to cross the blood–brain barrier and reflect the pathophysiology of brain injury. In this study, we aimed to investigate the role of salivary exosomal biomarkers in mTBI and PCS. Methods: A systematic review using the PRISMA guidelines was conducted, and studies were selected based on their relevance to the topic. Results: The analyzed studies have shown that exosomal tau, phosphorylated tau (p-tau), amyloid beta (Aβ), and microRNAs (miRNAs) are potential biomarkers for mTBI and PCS. Specifically, elevated levels of exosomal tau and p-tau have been associated with mTBI and PCS as well as repetitive mTBI. Dysregulated exosomal miRNAs have also been observed in individuals with mTBI and PCS. Additionally, exosomal Prion cellular protein (PRPc), coagulation factor XIII (XIIIa), synaptogyrin-3, IL-6, and aquaporins have been identified as promising biomarkers for mTBI and PCS. Conclusion: Salivary exosomal biomarkers have the potential to serve as non-invasive and easily accessible diagnostic and prognostic tools for mTBI and PCS. Further studies are needed to validate these biomarkers and develop standardized protocols for their use in clinical settings. Salivary exosomal biomarkers can improve the diagnosis, monitoring, and treatment of mTBI and PCS, leading to improved patient outcomes.

## 1. Introduction

A traumatic brain injury (TBI) is often caused by a blunt head trauma event significantly affecting the quality of life and health of an individual. In some cases, a TBI can even threaten a patient’s life, as it has been shown that TBI is associated with increased morbidity and mortality rates worldwide. Mild TBI (mTBI), also known as a concussion, accounts for approximately 80% of all TBI cases [1]. Despite the generally favorable outcomes for most mTBI patients, some individuals continue to experience chronic post-concussion (PC) symptoms, including cognitive impairment, headaches, sleep disturbances, and mood disorders, which can significantly impact their quality of life [2]. Early diagnosis and management of mTBI and its associated post-concussion syndrome (PCS) are crucial to prevent long-term complications. However, the current diagnostic methods for mTBI and PCS are limited and often rely on subjective clinical evaluations [2].

Exosomes are small extracellular vesicles released by almost all mammalian cell types, including neurons and glial cells [3]. Neuron-derived exosomes are extracellular vesicles released by neurons. They are thought to be key mediators in communication and waste management within brain tissues [4]. The diameters of exosomes vary between 30 and 150 nm [5]. Exosomes were first described in the 1980s [6]. They were thought to originate from the endomembrane system, while their membranous envelope is invaginated during the maturation process and forms the intraluminal vesicles (ILVs).

The ILVs consist of proteins, nucleic acids, and lipids. Mature endosomes that contain numerous ILVs are called multivesicular bodies (MVB) [7]. Multivesicular bodies are either degraded by lysosomes or transported to cell membranes; fuse with the cell membranes; and release the inner vesicles into the extracellular space, forming exosomes loaded with proteins, non-coding RNAs, lipids, and other active substances [8]. They contain particular and varied types of markers that contribute to identifying their origins. Once they are secreted, they can be internalized by recipient cells through different mechanisms, such as phagocytosis, micropinocytosis, endocytosis, and plasma membrane fusion [9,10,11]. Neuron- and glial-derived exosomes carry and release multiple molecules related to neuronal function and neurotransmission in the brain. They are essential in neuronal development, neuroimmune communication, and synaptic spasticity [11].

The roles of exosomes and the changes in the exosomal content in TBI have been extensively investigated over the past few years. Changes in the levels of exosomal content after a TBI can assist in the diagnosis and severity classification of the TBI in question [12]. The concentration of neuro-derived exosomes in the plasma of patients with an mTBI is reduced by 45% in the acute phase of the injury, while alterations in the levels of neuropathological protein in these exosomes can reveal phase and severity specificity [13]. Moreover, it has been shown that a pattern in exosomal content dynamics can be seen during different periods after an mTBI [14]. In this way, several proteins can be qualitatively and quantitatively assessed in the plasma of mTBI patients to prognose remote, long-term symptoms and recovery, as reported by Guedes et al. for plasma exosomal neurofilament light-chain peptide (NfL) levels [15]. The protective role of exosomes in TBI has also been documented. More specifically, it has been shown that exosomal miRNAs can improve neurodegeneration following repetitive mTBIs [16], suppress the inflammatory process, promote axonal growth, and improve neuroprognosis [17,18]. Recent studies have identified potential biomarkers, including microRNAs, as well as tau protein, cytokines, and other proteins, that could assist in mTBI and PCS diagnosis and management.

We aimed to describe the role of salivary exosomal biomarkers in mTBI and PCS and discuss their potential as diagnostic and prognostic tools based on a systematic review of the recent literature.

## 2. Materials and Methods

### 2.1. Literature Search

A comprehensive literature search was performed using PubMed, Embase, and Scopus databases to identify articles published until September 2022. The following search terms were used: “mild traumatic brain injury”, “concussion”, “post-concussion syndrome”, “exosomes”, “biomarkers”, “proteomics”, “microRNAs”, and “RNA-seq”. The search was limited to studies conducted on human patients and published in English. PRISMA guidelines were followed (PRISMA checklist attached as Appendix A). The systematic review was registered in PROSPERO platform under the ID 478729/1.11.2023.

### 2.2. Inclusion and Exclusion Criteria

Studies that investigated the role of exosomal biomarkers in mTBI and PCS were included. Studies that did not report data on salivary exosomes or biomarkers, case reports, and animal studies were excluded.

### 2.3. Data Extraction

Two authors independently carried out data screening and selection according to PRISMA guidelines. The eligibility criteria were then applied to the full texts of the studies. Data were extracted from the included studies using a standardized data extraction form that included study design, participant characteristics, types of biomarkers measured, and main findings.

### 2.4. Quality Assessment

The quality of the included studies was assessed using the Cochrane Risk of Bias Tool for randomized controlled trials and the Newcastle–Ottawa Scale for observational studies. Disagreements regarding data selection and quality assessments, if any, were resolved via carried out discussion until reaching consensus.

### 2.5. Data Synthesis

A narrative synthesis approach was used to summarize the findings of the included studies. Data were organized by the types of the measured biomarkers and their potential roles in diagnosis or prognosis.

## 3. Results

While the initial search yielded 65 studies, duplicate elimination led to the inclusion of 37 studies in the screening process. After the initial screening process, 22 additional studies were removed due to lack of availability of the full texts. Then, the full texts of the remaining 15 studies were screened. Five studies were excluded as they did not present relevant data, and, finally, ten studies were included in the present study (Figure 1). There were no concerns regarding the quality of the studies that were finally included (Figure 2).

The risk of bias was evaluated across ten domains (D1 to D10), which are central to the methodological rigor of clinical research. The bar chart (Figure 2) reveals that the majority of the studies exhibit a low risk of bias across most domains. This suggests that the studies generally employed robust methodological standards, enhancing the credibility of their findings. However, there remains a non-negligible proportion of domains where some concerns have been noted, indicating areas where the studies could be prone to bias, albeit not of a degree sufficient to categorize them as high risk.

The heat map (Figure 2) provides a detailed visualization of the risk of bias on a study-by-study basis. It allows for identifying patterns within and across studies, offering a granular view that is not immediately apparent in the aggregate data. Certain studies demonstrate a consistent low risk of bias across all domains, bolstering the reliability of their contributions to the field. Conversely, other studies exhibit variable levels of risk across domains, necessitating a cautious interpretation of their findings.

Domain D1 consistently shows a low risk of bias, suggesting that the initial design and conceptual framing of the corresponding studies are generally sound. Domains D5 and D10, however, show a higher incidence of ‘Some concerns,’ which may reflect challenges in the operational aspects of study execution or specific methodological weaknesses. It is crucial for future research to address these concerns to ensure the generation of high-quality evidence.

The overall low risk of bias enhances the strength of the systematic review’s conclusions, providing a solid foundation for clinical recommendations and further research. Nevertheless, the presence of some concerns across certain domains and studies underscores the need for continuous methodological vigilance.

### 3.1. Narrative Summary of the Studies

Kenney et al. measured plasma and exosomal levels of tau protein, p-tau, and amyloid beta peptide (Aβ) in Veterans with a history of mTBI and chronic neuropsychological symptoms [19]. They reported that exosomal p-tau and tau peaks were associated with repetitive mTBIs and correlated with PC symptoms, suggesting that blood-based exosomes could provide peripherally sourced information about the effects of mTBI on brain tissues.

Wang et al. focused on developing a blood-based biomarker assay for mTBI using circulating exosomes and showed that the analysis of circulating exosomes via acoustofluidic exosome separation allows the detection of exosomal biomarkers in TBI [20]. Also, the study pointed out that these changes could predicted in the first 24 h following head trauma, potentially constituting a considerable advantage. Thus, acoustofluidic exosome separation could improve early diagnosis and treatment decisions associated with TBI.

Devoto et al. conducted a study to investigate the potential role of exosomal microRNA expression in neurobehavioral outcomes among Veterans with mTBI caused by blunt-force or blast injuries [21]. The preliminary results originating from 152 participants described several dysregulated cellular pathways involved in neurodegeneration, inflammation, and central hormonal regulation that could be related to chronic neurobehavioral symptoms after a blast-induced TBI. The same group further investigated the role of exosomal micro-RNAs (miRNAs) in relation to the persistence of chronic TBI symptoms [22]. In this longitudinal study, dysregulated exosomal miRNAs were identified in participants with a history of mTBI, particularly those with repetitive mTBIs, correlating exosomal miRNAs with inflammatory regulation, neurological disease, and cell development. Thus, exosomal miRNAs analysis could also provide novel insights into the underlying pathobiology of chronic TBI symptom persistence.

Gill et al. also conducted a study to identify biomarkers in peripheral blood related to chronic post-concussive and behavioral symptoms following TBI [23]. Their study investigated the concentrations of tau, amyloid β 42 peptides, and cytokines (TNF-α, IL-6, IL-10) in neuron-derived exosomes from the peripheral blood of military personnel with or without mTBI, concluding that increased exosomal tau, amyloid β 42, and IL-10 concentrations could be found in the plasma of mTBI patients compared to the controls. However, a relationship between PC symptoms and exosomal tau concentration increases was not reported, yet exosomal IL-10 concentrations correlated with post-traumatic stress disorder (PTSD) symptoms.

Goetzl et al. discussed the potential mechanisms and predictive biomarkers for chronic traumatic encephalopathy (CTE) after acute mTBI [24]. Their promising hypothesis explained a possible pathway for the progression of PCS to CTE that included changes occurring in the physiological neuronal proteins, such as prion cellular protein (PRPc), coagulation factor XIII (XIIIa), synaptogyrin-3, IL-6, and aquaporins. They found that in patients with long-term symptoms occurring after an mTBI, the levels of neuron-derived exosome concentrations in some biological fluids could be maintained for several months. This could suggest that neuron-derived exosomes could be significant contributors to CTE development, yet the pathological mechanisms through which they act are not clearly understood, and they are thought to take direct actions or to interact with amyloid β peptides or p-tau proteins. To address this aspect, Goetzl et al. investigated the levels of neuron-derived exosomes and their plasma transporters in individuals with or without acute and chronic mTBI [14] and found that exosomal levels of brain functional proteins were significantly abnormal in cases of acute mTBI. By contrast, chronic mTBI was characterized by normal levels of exosomal brain functional proteins, but exosomal brain pathogenic protein levels were elevated. This could mean that some exosomal proteins could be promising biomarker candidates for both acute and chronic mTBI and potentially predictive of the long-term effects of mTBI as well as mTBI-induced neurodegeneration.

Further research on the potential of neuronally derived exosomes and astrocyte-derived exosome cargo proteins as biomarkers of chronic mTBI was conducted by Winston et al. in relation to military service members [25]. They found that plasma neuron-derived and astrocyte-derived exosomal amyloid β 42 peptide levels were significantly increased in patients affected by chronic mTBI. By contrast, plasma neuron-derived and astrocyte-derived exosomal neurogranin levels were found to be significantly decreased in mTBI patients compared to healthy controls. These findings could suggest that neuron-derived exosomes could be used to describe the pathomechanisms of TBI.

Regarding the potential of plasmatic exosomal content in predicting PC symptoms’ chronicity, Guedes et al. investigated veterans with a history of mTBI and found that repetitive mTBIs as well as chronic PCS were characterized by increased plasma and exosomal levels of NfL. Furthermore, they reported that these molecules could be also associated with the presence of several symptoms of PTSD, such as depression [15]. Based on the molecular mechanisms underlying mTBI and the changes they reported in exosomal content, they proposed that the persistence of PCS could be tied to neuroinflammatory and disruptive impairments. Following a similar hypothesis, Meier et al. found that some extracellular-vesicle-associated cytokines levels could be predictive of the duration of PC symptoms [26]. In this way, IL-6 originating from extracellular vesicles was reported to follow this pattern and to significantly correlate with the period that passed since the mTBI and the plasmatic profile dynamics, leading to the conclusion that extracellular-vesicle-associated IL-6 could be an important biomarker for concussions. Another study conducted by Guedes and colleagues investigating the correlation between PTSD symptom severity and the peripheral levels of extracellular vesicle proteins and miRNAs in chronic mTBI reported that extracellular vesicles’ NfL levels were significantly increased in patients with a history of mTBI and were corelated with more-severe PTSD symptoms as compared to healthy controls and, for mTBI patients, with less-severe PTSD symptoms [27]. These findings could suggest that levels of NfL could predict the severity of PTSD symptoms in individuals with a history of mTBI.

### 3.2. Tau Protein

Exosomal tau protein was identified as a potential biomarker for mTBI and PCS. Increased levels of exosomal tau were observed in individuals with a history of repeated mTBIs compared to those with fewer or no mTBIs [25], whereas a significant correlation between exosomal tau levels and post-traumatic and PC symptoms, which could include cognitive, affective, and somatic symptoms, was reported. Accordingly, it was suggested that peripheral levels of exosomal tau could provide relevant information about the effects of mTBIs on brain tissues due to its contribution to the mechanisms underlying the chronic feature of the specific neurophysiological symptoms.

On the other hand, phosphorylated tau protein levels were reported to be significantly increased in exosomes isolated from plasma samples of individuals with mTBI and correlated with repetitive head trauma events [19,25,26,27]. Furthermore, these changes were associated with the chronic neuropsychological symptoms of PCS. Therefore, exosomal phosphorylated tau could be considered a potential biomarker for mTBI and PCS and may also be able to provide insights into the underlying pathobiology of these conditions.

### 3.3. Exosomal miRNAs

MiRNAs are small non-coding RNAs that play a role in post-transcriptional gene regulation. Two studies investigated miRNAs’ roles as potential biomarkers for mTBI and PCS, as well as with regard to their potential involvement in the underlying pathobiology of these conditions [22,27]. Some specific miRNAs that are dysregulated in exosomes were found to contribute to intercellular communication and to be significantly associated with mTBI and PCS symptoms’ severity and duration. In this context, Devoto et al. [22] reported that more than 20 miRNAs were dysregulated when comparing mTBI patients with controls. They also pointed out that there could be significant differences in brain tissues’ responses to the different types of head injury, as they found that the expression of 23 miRNAs was significantly different in blast versus blunt mTBI. Furthermore, Guedes et al. [27] reported that miR-139–5p, miR-204–5p, miR-372–3p, miR-509-3–5p, miR-615–5p, and miR-1277–3p expressions were different in mTBI patients versus healthy controls and that four other miRNAs have significantly different expression in mTBI patients that exhibit PTSD symptoms. Thus, the dysregulated miRNAs were analyzed in the context of their involvement in various pathways, including inflammation, neuronal repair, and cellular development.

### 3.4. Exosomal Cytokines

The role of exosomal cytokines as biomarkers for mTBI and PCS was also investigated, and the levels of IL-6 and IL-10 were reported to be significantly increased in neuron-derived exosomes isolated from the peripheral blood of military personnel with mTBI [23]. Also, post-injury extracellular-vesicle-associated IL-6 levels were significantly increased and positively associated with the number of days injured athletes reported PC symptoms [26]. Similar findings were reported by Goetzl et al., but they further reported that impaired IL-6 levels could also be correlated with the chronic phase of mTBI and PCS and possibly contribute to the development of CTE [24].

### 3.5. Neuron-Derived Exosomes

The potential role of neuron-derived exosomes in transporting proteins and neurotoxic forms of amyloid β peptides or p-tau in the context of CTE was investigated by Goetzl et al. [24]. They described several pathways through which neuron-derived exosomal content (specifically IL-6 and aquaporins) could contribute to persistent PCS, neuroinflammation, and eventually CTE development (Table 1). The pathophysiological mechanisms of CTE were also described in the context of neuron-derived exosomal content isolation from peripheral blood, as PRPc, XIIIa, and synaptogyrin-3 were found to interact with neurotoxic molecules within brain tissues and act as transporters sending cargo to peripheral regions.

## 4. Discussion

Mild TBI and PCS are complex disorders with a range of symptoms that can persist long after the initial injury [1,2]. Our previous studies described persistent PCS symptoms and their implication in patients’ recovery following an mTBI event [28]. Also, we found that the persistent symptomatology could be associated with brain volumetric changes and long-term brain molecular imbalance [29]. However, there might be suggestive molecular changes that could be addressing a certain window in time following a traumatic event [30]. In this context, the lack of reliable biomarkers for mTBI and PCS has hindered these disorders’ diagnosis, comprehension, and treatment [2]. Our studies previously suggested the need for biomarkers that can be found in easily obtainable biological fluids. In a recent review, we described the potential of saliva to contain several dozens of molecules with critical relevance for mTBI [30]. Not only is saliva far easier to obtain compared to other biological fluids and preferred by patients, but it is also a possible source allowing for the faster detection of biochemical changes after traumatic events [30]. Nevertheless, mTBI is not the sole pathological event that leads to changes in the exosomal content of the saliva, as Zhang et al. [7] and Hoffman et al. [31] previously documented significant differences between healthy and cancer patients’ salivary exosome profiles.

Recent studies have shown promising results with respect to using salivary exosomal biomarkers as potential diagnostic and prognostic tools for mTBI and PCS. Also, the importance of exosomes was previously described in relation to mTBI and PCS, and studies have shown that their detection could be significantly correlated with the time passed after a traumatic event and the extent of the subsequent damage [32]. Naturally, the role of exosomes within the brain tissues is to mediate intercellular communication [33]. However, in cases of a crisis, exosomes were found to act as potent mediators of neuronal response to stress, inflammation, and regeneration [34,35]. Furthermore, exosomes are potent molecular complexes that can be successfully isolated or even synthetized in vitro and used as potential therapeutic agents [36]. They can, moreover, be efficient carriers of active therapeutic biomolecules [37]. In this context, various therapeutic applications have recently been described in regenerative medicine, showing great potential in neurodegenerative disease treatment [38].

One of the most-studied salivary exosomal biomarkers for mTBI and PCS is tau protein, a microtubule-associated protein stabilizing neuronal axons. Tau protein is also implicated in microtubule-mediated axonal transport, making it a key player in neuronal development [39]. However, the active implication of tau aggregation in predisposing one to tauopathies was previously demonstrated only for moderate to severe TBIs [40]. Despite this, it was shown that increased levels of tau protein within biological fluids are mainly present within the first 24 h post-TBI [41]. The balance between tau and its active phosphorylated form is a known biomarker for acute and chronic TBIs [42]. In several of the studies discussed herein, increased levels of tau protein originating from exosomes were found in the saliva of individuals with mTBI and PCS, indicating axonal damage and neuronal degeneration [19,25,26,27]. Additionally, exosomal phosphorylated tau was found to be elevated in the saliva of individuals with repeated mTBIs, suggesting a potential link between repetitive mTBIs and chronic neurodegenerative disorders, such as CTE. As tau protein levels increase in biological sources obtained from patients with Alzheimer’s disease (as repeatedly reported [43,44]), older studies report unclear correlations between mTBI and Alzheimer’s disease. However, more recent studies showed that mTBI could predispose one to AD and related dementias, regardless of age or sex [45,46].

Exosomal miRNAs are another class of biomolecules that have shown promise as salivary biomarkers for mTBI and PCS, as several studies discussed herein found dysregulated levels of exosomal miRNAs in the saliva of individuals with mTBI and PCS [22,27]. MiRNAs are short RNA molecules consisting of non-coding sequences that regulate gene expression. Some of the miRNAs that were isolated from post-mTBI patients were associated with dendritic differentiation and synaptic function [32]. One study suggested that dysregulated exosomal miRNAs could be associated with inflammation and neuronal repair pathways in individuals with repeated mTBI [22]. Exosomal miRNAs were also associated with neurological disease, developmental injury and abnormalities, and neuropsychiatric disease, as well as with chronic mTBI [27]. While several miRNAs were shown to be implicated in mTBI and PCS, as changes in biological fluids levels were previously documented, they were also described as potential therapeutic targets in both animal and patient studies [32].

Cytokines, such as IL-6 and IL-10, have also been investigated as potential salivary exosomal biomarkers for mTBI and PCS, and exosomal TNF-α, IL-6, and IL-10 levels were observed to be significantly increased in individuals with mTBI, as compared to healthy controls [23,24,26]. The roles of cytokines are mainly tied to the inflammatory response; however, in the brain, the activity of cytokines was also described as modulatory in pathways such as learning and memory, neuronal development and differentiation, synaptic plasticity, the blood–brain barrier, and sleep regulation [47]. Recent studies have investigated the role of salivary exosomal PRPc, XIIIa, synaptogyrin-3, IL-6, and aquaporins in mTBI and PCS and reported that the prolonged increased levels of aquaporins and IL-6 in neuron-derived exosomes might contribute to the persistent central nervous system oedema and inflammation observed in CTE [24]. In this context, the molecular dysregulations caused by mTBI target brain circulation and blood coagulation, brain water balance and edema formation, the tau-accumulation-associated signaling pathway, and the acute-phase inflammatory response. On the other hand, counteracting measures have been described to be designed for homeostasis and repair as well as the replacement of the damaged cells [48,49]. A recent animal model study showed that Il-6 and TGF-β are implicated in macrophage infiltration and subsequent tissue repair [50]. In this context, it would be interesting to study the potential of exosomal cytokines in regeneration after an mTBI event. Thus, future research could focus on describing the changes in exosomal miRNAs’ expression in correlation with mTBI-affected brain molecular pathways to further uncover the altered signaling pathways that lead to mTBI and PCS symptoms and outcomes, as well as possible means of overcoming them and preventing long-term effects.

Overall, the discussed studies suggested that salivary exosomal biomarkers, including tau protein, miRNAs, and several cytokines, could be promising diagnostic and prognostic tools for mTBI and PCS. These biomarkers can provide insight into the underlying pathomechanisms of these disorders as well as assist in the development of targeted therapies for mTBI and PCS. In this way, the importance of this study lies in the perspective we aimed to focus on regarding the potential of salivary mTBI biomarkers, in contrast to other recent studies that were mainly focused on blood biomarkers. However, further research is needed to fully understand the roles of these salivary exosomal proteins in the context of mTBI and PCS and validate these biomarkers according to their clinical utility.

## 5. Conclusions

In conclusion, using exosomal biomarkers in saliva has shown great promise in diagnosing and managing mTBI and PCS. There are several studies identifying several exosomal biomarkers for mTBI and PCS, namely, tau protein, p-tau protein, miRNAs, IL-6, and IL-10. These salivary biomarkers could provide a non-invasive diagnostic tool and assist in identifying individuals at risk of developing chronic symptoms and progression to CTE. Further research could focus on exosomal biomarkers, as they may provide valuable insight into the underlying pathophysiology of mTBI and PCS that could lead to targeted therapies. Furthermore, future studies are needed to validate these biomarkers in larger cohorts, determine their sensitivity and specificity in diagnosing mTBI and PCS, and aid in the discovery of reliable ways of separating salivary exosomes and quantifying exosomal content. Overall, exosomal biomarkers in saliva show great potential in the diagnosis and management of mTBI and PCS, and future research in this area can significantly improve patient outcomes.

## Figures and Tables

**Figure 1 jpm-14-00035-f001:**
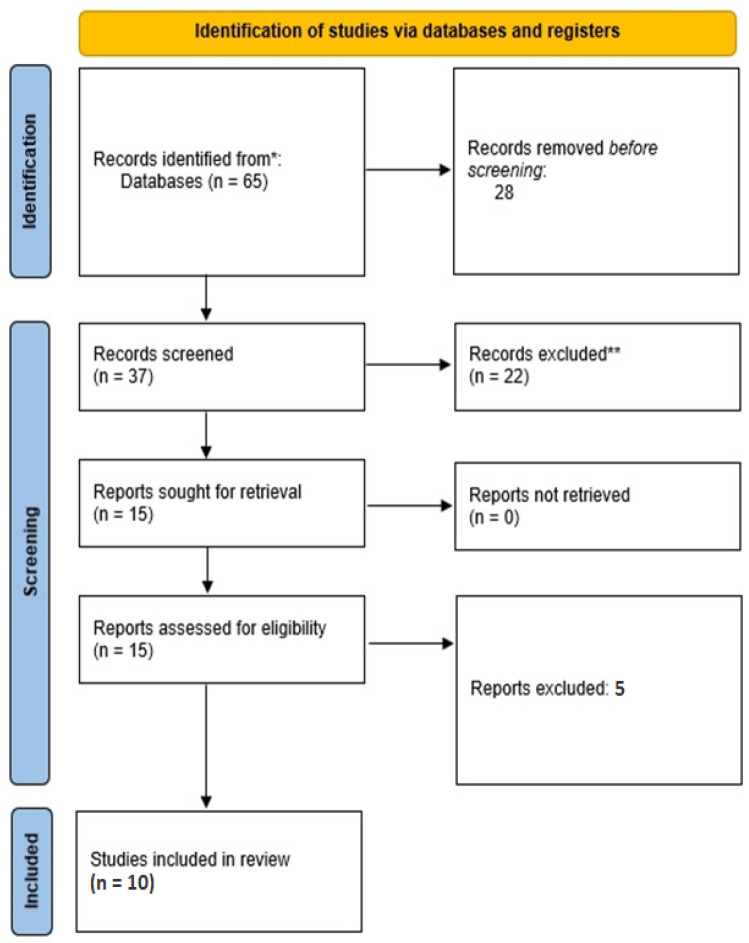
PRISMA flowchart of the study regarding the potential of exosomal biomarkers with regard to mild traumatic brain injury and post-concussion syndrome. * Databases searched online Pubmed, Web of Science and Scopus. ** Exclusion criteria: studies that did not report data on salivary exosomes or biomarkers, case reports, and animal studies.

**Figure 2 jpm-14-00035-f002:**
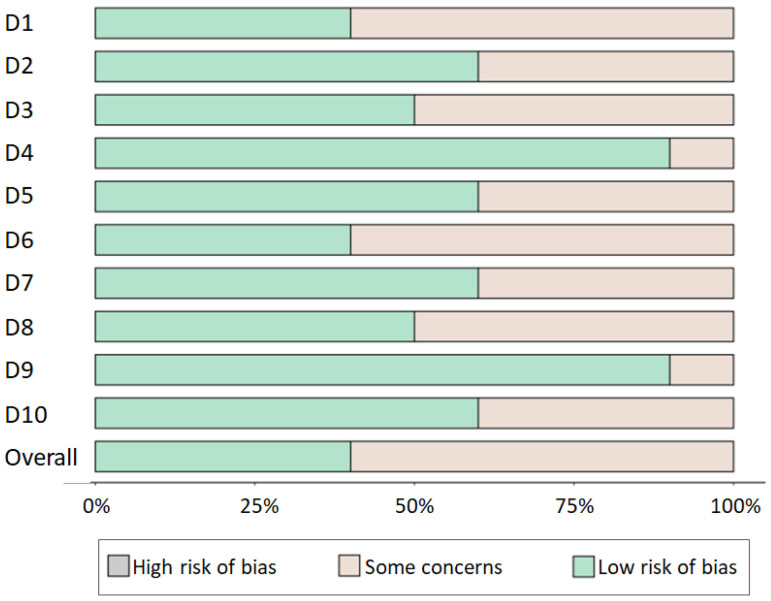
Quality assessment for the studies included in this systematic review. The risk of bias within the included studies was evaluated across ten domains, each reflecting a potential source of systematic error that could impact study findings. These domains were carefully selected to encompass the breadth of factors contributing to clinical research’s integrity and validity. D1—Selection Bias, D2—Performance Bias, D3—Detection Bias, D4—Attrition Bias, D5—Reporting Bias, D6—Publication Bias, D7—Confounding Bias, D8—Sampling Bias, D9—Measurement Bias, and D10—Other Biases [15,19,20,21,22,23,24,25,26,27].

**Table 1 jpm-14-00035-t001:** Recent highlights regarding the role of TBI biomarkers.

Exosomal Biomarkers	Recent Findings	References
tau and p-tau	Both exosomal tau and p-tau level changes were reported in mTBI and repetitive mTBI; exosomal p-tau levels were correlated with chronic neurophychological symptoms of PCS	[19,25,26,27]
micro-RNAs	The miRNAs for which changes in expression were reported in mTBI and PCS are involved in cell proliferation, cerebral microcirculation, apoptosis, tau accumulation, and neuronal differentiation	[22,27]
cytokines	IL-6 and IL-10 could be correlated with acute phase response in mTBI; IL-6 was associated with chronic post-concussion symptoms and the number of days of their manifestation	[23,24,26]
neuron-derived exosomes	Potential of transporting signal molecules, proteins (inflammatory factors), and neurotoxic peptides	[24]

## Data Availability

All data is available on request.

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
