# Peer review of "Exploring the Potential of Exosomal Biomarkers in Mild Traumatic Brain Injury and Post-Concussion Syndrome: A Systematic Review"

_jpm, 2023, doi:10.3390/jpm14010035_

Round 1
Reviewer 1 Report
Comments and Suggestions for Authors
The paper titled "Exploring the Potential of Exosomal Biomarkers in Mild Traumatic Brain Injury and Post-Concussion Syndrome: A Systematic Review" shows the authors' tremendous effort to summarize all available information about potential biomarkers of mTBI and PCS. The authors have done a commendable job in their research and analysis to show the role of salivary exosomal biomarkers.
Some parts of this paper should be improved:
Point 1. Please explain the figure 2 in detail. Authors should write explanations under the title of the figure.
Point 2. The authors explained ten studies according to the heat map in this review. According to Figure 1 and Results'text, 8 studies were analyzed. Please explain.
Point 3. In Alzheimer's disease, values of p-tau are also elevated. Do the patients with the mTBI and PCS had also Alzheimer's disease? Please include the response in the text of the paper.
Point 4. Throughout the text, there are some typos. Please correct these.
Author Response
Reviewer 1:
”The paper titled "Exploring the Potential of Exosomal Biomarkers in Mild Traumatic Brain Injury and Post-Concussion Syndrome: A Systematic Review" shows the authors' tremendous effort to summarize all available information about potential biomarkers of mTBI and PCS. The authors have done a commendable job in their research and analysis to show the role of salivary exosomal biomarkers.
Some parts of this paper should be improved:
Point 1. Please explain the figure 2 in detail. Authors should write explanations under the title of the figure.”
Response: Thank you for your kind words of appreciation and for your valuable suggestions. We added some explanations under the second figure and in the text.
”Point 2. The authors explained ten studies according to the heat map in this review. According to Figure 1 and Results'text, 8 studies were analyzed. Please explain.”
Response: We corrected the text and Figure 1, as the analysis was carried out on 10 studies. Thank you.
”Point 3. In Alzheimer's disease, values of p-tau are also elevated. Do the patients with the mTBI and PCS had also Alzheimer's disease? Please include the response in the text of the paper.”
Response: We found no mentions that the patients with mTBI and PCS in the selected studies had AD. However, tau protein changes were seen in many neurodegenerative diseases, such as tauopathies, AD, and chronic traumatic encephalopathy, the latter being known as the most prevalent long-term consequence of repeated mTBI. We added some information on the correlation between mTBI and AD in the Discussion section, as it was reported that repeated mTBI could predispose to AD later in life.
”Point 4. Throughout the text, there are some typos. Please correct these.”
Response: We revised the text and looked for the typos. Thank you for your kindness.
Reviewer 2 Report
Comments and Suggestions for Authors
In this work, the authors reviewed recent studies on exosomal biomarkers aiming to investigate roles of salivary exosomal biomarkers in mild traumatic brain injuries and post-concussion syndrome. The articles were selected following PRISMA guidelines and eight final studies were discussed in the manuscript regarding the potential of exosomal biomarkers as diagnosis and prognosis tools. The discussed studies suggested that salivary exosomal biomarkers, including tau protein, miRNAs, and several cytokines showed differential regulation after brain injuries. These conclusions suggest that these potential biomarkers could provide non-invasive diagnostic tools for mTBI and PCS. Overall the authors presented a comprehensive review on the exosomal biomarkers for mTBI and PCS, and the manuscript could be potential interest of JPM readers.
1. Initial search resulted in 65 studies and finally only 8 studies were included in the present work. The authors should give more clarities on the criteria of assessment and why many articles got excluded according to Fig 1.
2. The authors should increase the font size of Fig 2a and 2b. The information from Fig 2 is very hard to read.
Author Response
Reviewer 2:
”In this work, the authors reviewed recent studies on exosomal biomarkers aiming to investigate roles of salivary exosomal biomarkers in mild traumatic brain injuries and post-concussion syndrome. The articles were selected following PRISMA guidelines and eight final studies were discussed in the manuscript regarding the potential of exosomal biomarkers as diagnosis and prognosis tools. The discussed studies suggested that salivary exosomal biomarkers, including tau protein, miRNAs, and several cytokines showed differential regulation after brain injuries. These conclusions suggest that these potential biomarkers could provide non-invasive diagnostic tools for mTBI and PCS. Overall the authors presented a comprehensive review on the exosomal biomarkers for mTBI and PCS, and the manuscript could be potential interest of JPM readers.
- Initial search resulted in 65 studies and finally only 8 studies were included in the present work. The authors should give more clarities on the criteria of assessment and why many articles got excluded according to Fig 1.”
Response: We revised this part in order to be clearer. Some additional information was added at the beginning of the Results section.
”2. The authors should increase the font size of Fig 2a and 2b. The information from Fig 2 is very hard to read.”
Response: Figure 2 was revised for font size. Thank you for your kind corrections.
Round 2
Reviewer 1 Report
Comments and Suggestions for Authors
I am satisfied with this improved version.